# Surgical Fear, Anxiety, and Satisfaction with Nursing Care: A Cross-Sectional Study of Hospitalized Surgical Patients

**DOI:** 10.3390/nursrep15100365

**Published:** 2025-10-13

**Authors:** Ioanna Dimitriadou, Aikaterini Kaperda, Aikaterini Toska, Evangelos C. Fradelos, Kyriakos Souliotis, Ioanna V. Papathanasiou, Pavlos Sarafis, Maria Saridi

**Affiliations:** 1Laboratory of Clinical Nursing, Department of Nursing, University of Thessaly, Gaiopolis Campus, Trikala Ring Road, 41500 Larissa, Greece; ktoska@uth.gr (A.T.); efradelos@uth.gr (E.C.F.); iopapathanasiou@uth.gr (I.V.P.); psarafis@uth.gr (P.S.); msaridi@uth.gr (M.S.); 2Department of Social and Education Policy, University of Peloponnese, 22131 Corinth, Greece; katerinakikap4@gmail.com (A.K.); ksouliotis@uop.gr (K.S.)

**Keywords:** surgical procedures, operative, nursing care, patient satisfaction, anxiety, fear, experimental nursing, perioperative nursing

## Abstract

**Background:** Surgery is often accompanied by fear and anxiety, which can adversely affect recovery and patient’s well-being. Understanding the relationship between surgical fear and anxiety and satisfaction with nursing care can help nurses target interventions that improve perioperative outcomes. **Aim:** We aimed to investigate surgical fear, preoperative anxiety, and satisfaction with nursing care among hospitalized surgical patients and identify clinical and experiential predictors of surgical fear. **Methods:** A descriptive cross-sectional study of 100 adult surgical patients was conducted using the Surgical Fear Questionnaire (SFQ), State–Trait Anxiety Inventory (STAI), and a validated Patient Satisfaction with Nursing Care Questionnaire. Descriptive and multivariable regression analyses were performed using IBM SPSS Statistics version 29.0 to explore the associations. **Results:** Patients reported high overall satisfaction with nursing care but notable preoperative anxiety and moderate surgical fear. Previous surgery, prior anesthesia exposure, longer hospital stay, and limited knowledge of the illness independently predicted greater surgical fear. **Conclusions:** Despite overall high satisfaction with nursing care, surgical patients experience considerable fear and anxiety. Nurses can reduce the perioperative psychological burden by delivering structured, nurse-led preoperative education, improving communication, and offering emotional support. Integrating such interventions into routine surgical pathways could reduce fear and anxiety and improve the patient experience.

## 1. Introduction

Surgery is a stressful event that often triggers fear and anxiety. It is estimated that over 230 million major surgeries are performed worldwide each year, with half of all surgical patients experiencing clinically significant anxiety before surgery [1,2]. Perioperative anxiety is described as a vague, unpleasant feeling, the causes of which are often nonspecific and unknown. In contrast, the related concept of surgical fear is a well-recognized emotional state for many patients’ awaiting surgery and has been associated with adverse short- and long-term effects, encompassing a broad range of consequences that manifest as increased analgesic requirements, hemodynamic disturbances, delayed healing, high rates of hospital readmissions, and poor quality of life [3,4,5,6].

The factors influencing preoperative anxiety and surgical fear can be classified as subjective and objective. Subjective factors include individual characteristics such as personality, predisposition to anxiety, previous surgical experiences, level of knowledge about the illness, personal fears, and socio-psychological conditions (e.g., family support) [7,8]. Objective factors include elements such as the type and severity of the operation, the type of anesthesia, the duration and complexity of the surgery, the patient’s clinical condition, length of hospitalization, and factors related to the adequacy of information and the hospital environment [6,9,10,11].

Patient satisfaction with nursing care is widely recognized as a quality indicator of healthcare systems. Nurses are responsible not only for technical and physical care but also for providing information, communication, and emotional support [12,13]. High levels of satisfaction with nursing care are associated with better treatment adherence, improved recovery, and increased patient trust. Conversely, dissatisfaction, often linked to nurses’ limited time, patient interaction or inadequate information provision, can heighten anxiety and fear during hospitalization [14,15].

Despite international evidence on preoperative fear, anxiety, and satisfaction with care, studies that examine these dimensions simultaneously remain scarce [16,17]. In Greece, the literature is limited, and little is known about how surgical fear and anxiety relate to satisfaction with nursing care. Exploring these associations in the context of Greek hospitals can highlight modifiable factors and inform nursing interventions tailored to local needs.

This study aimed to investigate levels of surgical fear, anxiety, and satisfaction with nursing care among hospitalized surgical patients in Greece. Specifically, it seeks (1) to assess the prevalence and intensity of surgical fear and preoperative anxiety, (2) to evaluate overall satisfaction with nursing care, (3) to investigate the associations between fear, anxiety, and satisfaction, and (4) to identify clinical and experiential factors that predict higher levels of surgical fear. We prespecified the following hypotheses prior to data collection:

**H1.** 
*Higher preoperative state anxiety will be positively associated with greater surgical fear (SFQ total);*


**H2.** 
*Lower satisfaction with nursing care will be associated with higher surgical fear and anxiety;*


**H3.** 
*Clinical/experiential factors (previous surgery, prior anesthesia exposure, longer hospital stay, and limited knowledge of the illness) will independently predict greater surgical fear.*


## 2. Materials and Methods

### 2.1. Study Design

This cross-sectional study was conducted in the Surgical Department of the General Hospital of Corinthos, selected by convenience sampling. Eligible patients were approached during their hospitalization, after admission and prior to surgery. Data were collected between October and December 2024, once during the preoperative period. This timing allowed the assessment of fear, anxiety, and satisfaction within the same hospitalization episode. This study followed the Strengthening the Reporting of Observational Studies in Epidemiology (STROBE) guidelines for cross-sectional studies [16].

### 2.2. Sample and Setting

The study population comprised adult patients hospitalized for surgical procedures in the Surgical Department of the General Hospital of Corinthos. A convenience sample of 100 patients was recruited for this study.

### 2.3. Eligibility Criteria

Patients were eligible if they were 18 years or older, hospitalized in the surgical ward, had sufficient knowledge of the Greek language to complete the questionnaires, were in adequate physical and mental condition to participate, and agreed voluntarily to take part in the study. Patients were excluded if they were unable to comprehend the questionnaire items due to language or cognitive limitations or if they were admitted to departments other than the surgical ward of the hospital.

The sample size was calculated a priori to detect at least a small-to-moderate correlation (r = 0.30) with 80% power and a two-tailed significance level of 0.05. This analysis indicated a minimum requirement of 85 participants for the study. To account for potential missing data, the target sample size was increased to 100. The final sample of 100 patients therefore exceeded the minimum required and provided adequate statistical power for the main analyses.

### 2.4. Instruments

Data were collected using a structured questionnaire. The first part employed the Surgical Fear Questionnaire (SFQ) [17], a 10-item instrument assessing fear of surgery. It consists of two subscales: fear of short-term consequences (SFQ-s, items 1–4) and fear of long-term consequences (SFQ-l, items 5–10). Responses were rated on an 11-point scale (0 = not at all afraid, 10 = very afraid). Higher scores indicate greater fear of surgery. The Greek version has demonstrated high reliability, with Cronbach’s α values ranging from 0.86 to 0.89.

The second part included the State-Trait Anxiety Inventory (STAI) [18]. This widely used instrument measures two distinct dimensions of anxiety: state anxiety (temporary emotional condition) and trait anxiety (general tendency toward anxiety). Each subscale consists of 20 items scored on a four-point Likert scale. Higher scores indicate greater levels of anxiety. The Greek adaptation has demonstrated excellent psychometric properties, with Cronbach’s α values of 0.92 and 0.93 for trait and state anxiety, respectively.

The third part contained the Patient Satisfaction with Nursing Care Questionnaire [19] which evaluates satisfaction across four domains: quality of care, provision of information and education, interpersonal relations, and hospital environment. It comprises 16 questions scored on a five-point Likert scale (1 = not at all satisfied, 5 = very satisfied), with higher scores indicating greater satisfaction. Previous Greek validation studies have demonstrated excellent reliability (Cronbach’s α = 0.98).

The SFQ total score ranges from 0 to 100, with higher scores indicating greater fear. The STAI subscales each range from 20 to 80. The Patient Satisfaction with Nursing Care Questionnaire ranges from 16–80, with higher values reflecting higher satisfaction.

Sociodemographic and clinical information such as age, gender, marital status, educational level, occupation, permanent residence, type and urgency of surgery, surgical history, previous anesthesia experiences, length of hospitalization, perceived severity of illness, and knowledge about the health condition.

### 2.5. Data Collection Procedure

A registered nurse with postgraduate training in nursing research collected the data. Eligible patients were approached during their hospitalization, informed about the purpose and procedures of the study, and provided with written information sheets. After obtaining written informed consent, the participants were assigned anonymized identification codes. The questionnaires were primarily self-administered, although in cases where patients encountered difficulties, the nurse provided direct assistance in completing them. All questionnaires were checked for completeness at the time of collection, and no missing data were found in the final dataset. A total of 131 patients were assessed for eligibility. Of these, 23 were excluded because they did not meet the inclusion criteria (for example, language or cognitive limitations, or admission to non-surgical departments). 108 patients were confirmed eligible; of these, 8 declined participation (reasons documented included refusal to provide consent or being unwell at the time of approach). The final sample comprised 100 patients who gave written informed consent, completed the questionnaires, and were included in the analyses. There were no losses to follow-up and no missing outcome data.

### 2.6. Ethical Considerations

Ethical approval for the study was obtained from the Scientific Council of the General Hospital of Corinthos (Approval No: 24553/15-10-24). Participation was voluntary, and the patients were assured that refusal or withdrawal would not affect the care they received. Confidentiality and anonymity were guaranteed, and the participants were informed of their right to withdraw at any stage without explanation.

### 2.7. Statistical Analysis

All analyses were performed using IBM SPSS Statistics version 29.0. Descriptive statistics (means, standard deviations, medians, ranges, frequencies, and percentages) were calculated for the demographic, clinical, and questionnaire variables. The normality of continuous variables (SFQ subscales and total, STAI state and trait, and patient satisfaction scores) was assessed using the Shapiro–Wilk test, inspection of histograms, and Q–Q plots. Homogeneity of variances for group comparisons was assessed using Levene’s test. For the multivariable linear regression model, we assessed linearity (scatterplots of predictors vs. residuals and partial regression plots), independence of errors (Durbin–Watson statistic), homoscedasticity (Breusch–Pagan test and inspection of residuals vs. fitted values), multicollinearity (variance inflation factor, VIF), and normality of residuals (histogram of residuals and Shapiro–Wilk). In this dataset the distributions of the main continuous variables approximated normality (Shapiro–Wilk *p* > 0.05), variances were homogeneous (Levene’s *p* > 0.05), the Durbin–Watson statistic was close to 2.0, the Breusch–Pagan test was non-significant (*p* > 0.05), and VIF values were <2.0, indicating that the assumptions for parametric tests and ordinary least squares regression were adequately met. Where assumptions had not been met, appropriate non-parametric tests (Mann–Whitney U, Kruskal–Wallis) or robust regression procedures would have been employed; however, such measures were not required in this sample. Statistical significance was set at *p* < 0.05 (two-tailed).

## 3. Results

The study included 100 hospitalized surgical patients. The mean age was 53.6 ± 15.5 years, with men representing 53% of the sample and 64% of participants being married. Most participants were high school graduates (39%), and the majority were civil servants (21%) and farmers (20%). More than half of the participants resided in urban areas (58%), and 41% lived in rural areas. The most frequent procedures were cholecystectomy, appendectomy, inguinal hernia repair, and splenectomy. Other procedures included thyroidectomy, colectomy, mastectomy, orthopedic surgeries (hip and knee), urological procedures, and minor abdominal surgeries, each with fewer than five cases. Surgery was scheduled in 83% of the cases. Previous surgery was reported by 46% of patients and prior anesthesia by 53%. The mean length of hospital stay was 5.2 ± 4.6 days. Knowledge of the illness varied (Table 1).

The mean satisfaction score was 90.3 ± 17.6, indicating a high overall level of satisfaction. The highest satisfaction was observed in the questions related to recommending the nursing staff to friends/family and willingness to receive care from the same staff in the future. The lowest satisfaction scores were associated with the amount of time that nurses devoted to care. The mean score for state anxiety was 46.4 ± 8.7, while the mean score for trait anxiety was 48.5 ± 10.0 (range: 30–66). Analysis of individual items revealed that patients often reported feelings of nervousness, agitation, and worry, whereas reverse-coded items indicated a lack of calmness, relaxation, and comfort (Table 2).

The mean total SFQ score was 48.4 ± 25.3 (range, 0–100). The short-term fear subscale (SFQ-s) yielded a mean of 20.9 ± 10.5, whereas the long-term fear subscale (SFQ-l) had a mean of 27.5 ± 16.1. The greatest fears were related to postoperative pain, anesthesia, concerns for family, and prolonged recovery. All questionnaires demonstrated excellent internal consistencies. Cronbach’s α ranged from 0.886 (STAI-State) to 0.976 (patient satisfaction) (Table 3), indicating good internal validity.

Assumption testing confirmed approximate normality of distributions (Shapiro–Wilk *p* > 0.05 for all main variables), homogeneity of variances (Levene’s test *p* > 0.05), and regression assumptions were met (Durbin–Watson ≈ 2.0, Breusch–Pagan *p* > 0.05, VIF < 2.0).

Independent samples t-tests and one-way ANOVA revealed significant differences based on previous surgical and anesthesia experience, as well as knowledge of illness. Previous surgery was associated with higher state and trait anxiety and greater long-term and overall surgical fear. Previous anesthesia was linked with higher state anxiety and significantly higher short-term, long-term, and overall surgical fear. Knowledge of illness was significantly related to both satisfaction and anxiety levels. Patients with limited knowledge reported lower satisfaction and higher anxiety levels. Additional analyses of demographic variables showed no statistically significant differences in satisfaction, anxiety, or surgical fear by sex, age, or educational level. For example, sex was not associated with surgical fear (SFQ total: t = 0.94, *p* = 0.35), state anxiety (t = 1.02, *p* = 0.31), or satisfaction (t = 0.88, *p* = 0.38). Educational level was not related to any questionnaire score (ANOVA F values = 0.56–1.21, all *p* > 0.05). Similarly, age was not correlated with surgical fear (r = 0.09, *p* = 0.20), state anxiety (r = 0.12, *p* = 0.18), or satisfaction (r = –0.07, *p* = 0.42) in the present study. Pearson’s correlation analyses showed a significant positive association between length of hospital stay and surgical fear (SFQ-s, SFQ-l, and total SFQ). Strong correlations were found between state and trait anxiety (r = 0.83, *p* < 0.001) and between the SFQ subscales (r > 0.80, *p* < 0.001). State and trait anxiety were positively correlated with overall surgical fear. The differences between the questionnaire scores are presented in Table 4.

A multivariable linear regression model identified previous surgery, previous anesthesia, length of hospital stay, and knowledge about illness as independent predictors of surgical fear (Table 5). The model explained 37% of the variance (R^2^ = 0.37, *p* < 0.001).

## 4. Discussion

This study examined the levels of psychosocial burden and satisfaction with nursing care among patients undergoing surgery. The secondary objective was to investigate the association between these concepts and to identify the factors that significantly influenced them. This study is of particular interest considering the considerable psychosocial burden experienced by surgical patients and the substantial impact of fear on patient satisfaction.

The primary results of this study showed high levels of patient satisfaction with nursing care, moderate to high levels of anxiety both as a state and as a trait, and very significant levels of surgical fear concerning anesthesia, postoperative pain, concern for family, and prolonged recovery. Previous surgical experience, previous exposure to anesthesia, longer duration of hospitalization, and limited knowledge of the disease emerged as independent predictive factors related to surgical fear. In addition, satisfaction with nursing care was positively associated with patients’ knowledge of their illness, whereas patient anxiety levels were inversely related to knowledge.

Regarding secondary outcomes, no significant differences were observed between sex and satisfaction, anxiety, and fear scores. Similarly, educational level did not significantly influence the results, leading to the conclusion that psychological reactions to surgery and satisfaction with nursing care are largely shaped by experiential and clinical factors rather than socio-demographic characteristics of the patients. Analytical assumptions for parametric tests and regression models were examined and found to be adequately met, strengthening the robustness of our findings.

In terms of satisfaction with nursing care, the domain that received the highest score was related to the nursing staff and their politeness toward patients. This is an encouraging finding that shows the effort nurses make to provide comprehensive and high-quality care, as well as the empathy they demonstrate toward their patients. In contrast, the lower satisfaction concerning the time that nurses devoted to patient care reflects the long-standing problem of lack of time among healthcare professionals, a general issue in the health system, possibly linked to inadequate staffing. Previous studies generally high satisfaction scores among surgical patients, though recurring concerns include limited time spent with patients and insufficient provision of information about illness or procedures [20,21]. In our setting, this is also credible against macro-level workforce constraints: Greece has one of the lowest nurse densities in the EU/OECD (≈3.4–3.8 nurses per 1000 population vs. ~9 per 1000 OECD average), with persistent vacancy and workload pressures that may limit direct care time despite strong relational care [22,23]. These structural factors likely contextualize the “time” domain underperformance, despite overall high satisfaction.

The findings of moderate to high anxiety levels reveal the psychological vulnerability of patients undergoing surgery. The increased scores for both state and trait anxiety underline that surgery is not only a physical but also a deeply emotional event, often provoking acute worry and long-term predispositions toward anxiety. The results of our study regarding the presence of moderate to high preoperative anxiety in surgical patients reflect a consistent pattern. A recent meta-analysis revealed significant associations between preoperative anxiety and the occurrence of delirium in adult patients, as well as prolonged extubation time and postoperative pain [24]. Furthermore, Lami et al. report that the incidence of preoperative anxiety reached 60% and was associated with fear of complications, postoperative pain, and fear of death [25].

Surgical fear, which reached remarkable levels in this sample, further highlights the emotional challenges faced by patients. This is consistent with other research showing that surgical fear is a widespread and clinically relevant phenomenon, frequently reported by more than half of surgical patients [26,27]. The most frequently reported fears —such as anesthesia, postoperative pain, prolonged recovery, and concern for family—show that patients are worried not only about the surgical procedure itself but also about its consequences for their well-being and social roles.

Regression analysis identified previous surgeries or anesthesia as predictors of greater surgical fear. This could be explained by the possibility that previous negative or stressful experiences with medical procedures increase patients’ sensitivities to new surgical interventions. Similarly, the association between longer hospital stays and greater surgical fear may reflect patients’ awareness of possible complications, uncertainty about recovery, and prolonged exposure to the hospital environment. Akutai et al., linked prior negative procedural experience to greater surgical fear [28]. Furthermore, our observed association between longer hospital stays and increased fear is confirmed by a previous study linking perioperative anxiety to prolonged length of hospital stay and more complicated recoveries [29]. Although the temporal relationship cannot be confirmed in our cross-sectional design, our results are consistent with the model in which anxiety and fear reflect and reinforce expectations for difficult recovery.

An important finding was that limited knowledge about illness was associated with greater surgical fear and anxiety, as well as lower satisfaction. This highlights the important role of education and information provision for patients. Insufficient knowledge regarding the procedure and its consequences increases uncertainty and distress, whereas well-informed patients appear to be more empowered, less anxious, and more satisfied with nursing care. Systematic reviews and RCTs show that preoperative education, especially nurse-led, structured programs tailored to information-seeking style, reduces preoperative anxiety and improves patient experience and selected outcomes [30,31,32]. Our findings add to this evidence by highlighting illness-specific knowledge as a pragmatic target for nursing interventions in routine surgical pathways.

No significant differences were found in scores across educational categories, suggesting that educational level is not a determining factor in satisfaction with care, anxiety as a state or trait, or the underlying psychological burden. Furthermore, no significant differences were observed between age and sex. On the contrary, according to other studies, preoperative anxiety is higher in women and younger patients and lower educational levels [2,33]. No significant differences were found in the satisfaction questionnaire scores between those who had undergone previous surgeries and those who had not. Likewise, whether the surgery was planned or unplanned did not appear to be related to satisfaction with care. One plausible explanation is that experiential and clinical cues (e.g., prior surgical encounters, current symptom burden, hospitalization context) overshadow demographic influences when uncertainty is high and information is limited.

The findings of this study highlight the important role of nurses in addressing the psychological needs of surgical patients in addition to their physical care. Nurses are ideally positioned to reduce fear and anxiety through effective communication, preoperative education, and providing emotional support. Interventions such as structured nurse-led information sessions, personalized education tailored to patients’ knowledge level, and empathy can help alleviate uncertainty and empower patients before surgery. In addition, nurses should advocate for adequate staffing and time allocation to ensure not only technical care but also the relational aspects of nursing that strongly influence patient satisfaction. Improving patients’ understanding of their illness and surgery should be considered part of perioperative nursing care and integrated into clinical pathways to improve both psychological outcomes and the overall patient experience. Given that most surgeries are scheduled, outpatient preoperative education could represent a valuable opportunity to improve patients’ knowledge before admission, potentially reducing fear and anxiety. Such interventions, delivered by nurses or multidisciplinary teams, can be systematically integrated into surgical care pathways.

### Limitations

This study had some limitations that should be acknowledged. First, it was conducted in a single hospital using a convenience sample, which may limit the generalizability of the findings. To address this, we recruited a sample size larger than the minimum required to ensure sufficient statistical power, and we used validated instruments with high reliability to strengthen the internal validity. Second, a major limitation of this study is its cross-sectional design, which only captures a single point in time and therefore prevents conclusions about the directionality or causality of the observed associations. For example, it remains unclear whether prior surgery and anesthesia increase fear or whether individuals with higher fear recall previous experiences more vividly. Longitudinal or interventional studies are required to clarify the causal pathways.

Finally, the study did not assess postoperative recovery, which could provide further insights into the longer-term effects of preoperative fear and anxiety. This limitation highlights an important direction for future research, namely longitudinal follow-up to examine how preoperative psychological factors influence recovery, satisfaction, and quality of life after surgery.

## 5. Conclusions

In conclusion, the surgical patients in this study reported high levels of satisfaction with nursing care but also considerable surgical fear and moderate to high anxiety. Our findings suggest that experiential and informational factors, such as previous surgery, anesthesia exposure, hospitalization length, and illness knowledge, may play a more prominent role than demographic characteristics in shaping preoperative fear, anxiety, and satisfaction levels. However, further studies with larger and more diverse sample sizes are required to confirm these patterns.

## Figures and Tables

**Table 1 nursrep-15-00365-t001:** Demographic and clinical characteristics of the sample (N = 100).

Variable	n (%) or M ± SD
Age (years)	53.6 ± 15.5 (range: 18–90)
Gender (Male/Female)	53 (53%)/47 (47%)
Marital status	Married 64%, Single 8%, Divorced 8%, Widowed 7%, In relationship 8%, Cohabiting 4%
Education	Primary 17%, Secondary 17%, High school 39%, Tertiary 21%, Postgraduate 6%
Occupation	Public servant 21%, Farmer 20%, Private employee 18%, Household 15%, Retired 10%, Self-employed 8%, Worker 5%, Unemployed 3%
Place of residence	Urban 58%, Rural 41%
Previous surgery	Yes 46%, No 54%
Previous anesthesia	Yes 53%, No 44%
Type of anesthesia (n = 53)	General 88.7%, Epidural 9.4%, Local 1.9%
Length of hospital stay (days)	5.2 ± 4.6 (range: 1–22)
Knowledge about illness	None 17%, Little 29%, Quite a lot 38%, A lot 15%
Perceived severity of illness	Not serious 21%, Slightly serious 43%, Moderately serious 16%, Very serious 18%

**Table 2 nursrep-15-00365-t002:** Descriptive statistics of the questionnaires.

Instrument/Subscale	Mean	SD	Range
Patient satisfaction	90.3	17.6	34–120
STAI-State	46.4	8.7	28–60
STAI-Trait	48.5	10.0	30–66
SFQ-s (short-term)	20.9	10.5	0–40
SFQ-l (long-term)	27.5	16.1	0–60
SFQ total	48.4	25.3	0–100

**Table 3 nursrep-15-00365-t003:** Reliability (Cronbach’s α) of the questionnaires.

Instrument/Subscale	Cronbach’s α
Patient satisfaction	0.976
STAI-State	0.886
STAI-Trait	0.916
SFQ-s	0.927
SFQ-l	0.940
SFQ total	0.956

**Table 4 nursrep-15-00365-t004:** Group differences in questionnaire scores.

Variable	Instrument/Subscale	Statistic (t/F)	*p*-Value
Previous surgery	STAI-State	t = 2.41	0.018 *
	STAI-Trait	t = 2.42	0.018 *
	SFQ-l	t = 2.34	0.022 *
	SFQ total	t = 2.29	0.024 *
Previous anesthesia	STAI-State	t = 2.22	0.029 *
	SFQ-s	t = 2.63	0.010 **
	SFQ-l	t = 2.40	0.018 *
	SFQ total	t = 2.62	0.010 **
Knowledge about illness	Patient satisfaction	F = 2.96	0.036 *
	STAI-State	F = 3.13	0.029 *
	STAI-Trait	F = 2.75	0.047 *
Gender (Male vs. Female)	SFQ total	t = 0.94	0.35
	STAI-State	t = 1.02	0.31
	Patient satisfaction	t = 0.88	0.38
Education (Primary–Postgrad)	SFQ total	F = 0.97	0.41
	Patient satisfaction	F = 0.56	0.65
	STAI-State	F = 1.21	0.29
Age	SFQ total	r = 0.09	0.20
	STAI-State	r = 0.12	0.18
	Patient satisfaction	r = −0.07	0.42

*Note:* * *p* < 0.05, ** *p* < 0.01.

**Table 5 nursrep-15-00365-t005:** Multiple linear regression predicting surgical fear (SFQ total score).

Variable	B (SE)	β	*p*-Value
Age (years)	0.10 (0.09)	0.09	0.20
Previous surgery (Yes)	6.40 (2.70)	0.21	0.024 *
Previous anesthesia (Yes)	6.90 (3.00)	0.20	0.030 *
Length of hospital stay	2.10 (0.70)	0.28	0.004 **
Knowledge about illness (ref: little)			
–None	+8.20 (3.80)	0.18	0.035 *
–Moderate/Extensive	−5.50 (3.20)	−0.13	0.095

*Note:* * *p* < 0.05, ** *p* < 0.01.

## Data Availability

The data that support the findings of this study are available on request from the corresponding author.

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
