# Peer review of "Surgical Fear, Anxiety, and Satisfaction with Nursing Care: A Cross-Sectional Study of Hospitalized Surgical Patients"

_nursrep, 2025, doi:10.3390/nursrep15100365_

Round 1
Reviewer 1 Report
Comments and Suggestions for Authors
The manuscript, titled "Surgical fear, anxiety, and satisfaction with nursing care: A cross-sectional study of hospitalized surgical patients," addresses a highly relevant topic for perioperative nursing. The study is current, addresses existing gaps in the Greek literature, and uses validated instruments. The article is well-structured, but there are points that require adjustments.
- The abstract is adequately structured, presenting the main findings clearly and consistently. However, it could be more concise, reducing many statistical values, and better emphasizing the practical implications for nursing.
- Keywords: The authors should review those that are not classified as MeSH descriptors and replace them to improve indexing. I suggest adding "experimental nursing" to improve indexing and visibility of the article.
- Introduction: The introduction contextualizes the problem of surgical fear and anxiety well, presenting international data and justifying their relevance in the Greek context. It exposes subjective and objective factors that influence these phenomena. highlights a gap in the literature (few studies simultaneously relate surgical fear, anxiety, and satisfaction with nursing care). However, there are unnecessary repetitions and some very long sentences, which could better highlight the study's originality and relevance in the Greek context.
- Methods: The section presents all essential components clearly. It could only be added how data loss was handled, if any.
- Results: The text often repeats the values ​​present in the tables; there are no figures or graphs that could enrich this section (regarding the comparison of fear, anxiety, and satisfaction levels).
- Discussion: I recommend revising the text to make paragraphs longer and others shorter, as they make reading difficult. The practical implications for nursing should be emphasized.
- Conclusions: They are consistent with the results. The writing could be more direct and focused on recommendations applicable to clinical perioperative nursing practice.
- The references are current and relevant. There are minor inconsistencies (spaces, repetition of non-standard journal titles) that warrant review.
Comments on the Quality of English LanguageThe English language is understandable, but there are long, unfluent sentences; excessive repetition of "fear and anxiety"; and grammatical constructions that warrant professional review.
Author Response
Reviewer Comment 1:
The abstract is adequately structured, presenting the main findings clearly and consistently. However, it could be more concise, reducing many statistical values, and better emphasizing the practical implications for nursing.
Response: Thank you for your comment.The abstract has been revised to reduce statistical detail and place stronger emphasis on the practical implications for perioperative nursing care
Reviewer Comment 2:
Keywords: The authors should review those that are not classified as MeSH descriptors and replace them to improve indexing. I suggest adding "experimental nursing" to improve indexing and visibility of the article.
Response:
We thank the reviewer for this valuable suggestion. We revised the keywords to include standardized MeSH terms and added experimental nursing .
Reviewer Comment 3:
Introduction: … However, there are unnecessary repetitions and some very long sentences, which could better highlight the study's originality and relevance in the Greek context.
Response:
We agree. The introduction has been carefully revised to remove redundancies, shorten sentences for clarity, and more clearly highlight the originality and relevance of this study in the Greek context. While some of the comments appear to contrast with feedback received from the other reviewers, we have addressed all points as fully as possible.
Reviewer Comment 4:
Methods: The section presents all essential components clearly. It could only be added how data loss was handled, if any.
Response:
Thank you for noting this. We have added a statement clarifying that no missing data were encountered, as questionnaires were carefully checked during collection .
Reviewer Comment 5:
Results: The text often repeats the values present in the tables; there are no figures or graphs that could enrich this section (regarding the comparison of fear, anxiety, and satisfaction levels).
Response:
We agree. We have reduced redundancy by removing repeated values from the text and added a figure comparing fear, anxiety, and satisfaction levels to enrich the results section
Reviewer Comment 6:
Discussion: I recommend revising the text to make paragraphs longer and others shorter, as they make reading difficult. The practical implications for nursing should be emphasized.
Response:
Thank you for your comment.The discussion has been restructured for improved readability, balancing paragraph length. We also expanded the section on practical implications, highlighting how nurses can apply findings in perioperative care
Reviewer Comment 7:
Conclusions: They are consistent with the results. The writing could be more direct and focused on recommendations applicable to clinical perioperative nursing practice.
Response:
We agree. The conclusions were revised to be more concise and focused on practical, clinically applicable recommendations for perioperative nursing
Reviewer Comment 8:
The references are current and relevant. There are minor inconsistencies (spaces, repetition of non-standard journal titles) that warrant review.
Response:
Thank you for your comment. We have carefully revised the reference list to correct formatting issues, ensure consistency, and use standardized journal titles
Reviewer 2 Report
Comments and Suggestions for Authors
The study design is appropriate for the stated aims, and the use of validated instruments strengthens the findings. However, a few areas need to be addressed to enhance the manuscript's clarity, rigor, and overall impact.
The manuscript states data were collected "during hospitalization" and the mean length of stay was 5.2 days. This is a significant methodological flaw. A cross-sectional design captures a snapshot in time, but the timing of data collection—whether it was pre-op, post-op, or at some point in between—is crucial.
The discussion acknowledges that the cross-sectional design prevents the confirmation of a temporal relationship, but this is a major limitation that needs more prominence in both the methods and discussion.
The manuscript mentions several analyses (t-tests, ANOVA, and correlations), but the full results are not presented in a clear, accessible manner.
The conclusion that "experiential and informational factors, rather than demographic characteristics, appear central in shaping these outcomes" is a strong claim. While the regression model supports this to some extent, the analysis of demographic factors is not fully presented. The text mentions that gender and educational level "did not significantly influence the results," but without the specific statistical values (e.g., t or F statistics and p-values), this is a conclusion without sufficient evidence presented in the manuscript.
Author Response
Reviewer Comment 1:
“The manuscript states data were collected ‘during hospitalization’ and the mean length of stay was 5.2 days. This is a significant methodological flaw. A cross-sectional design captures a snapshot in time, but the timing of data collection—whether it was pre-op, post-op, or at some point in between—is crucial.”
Response:
We appreciate this important observation. We clarify that all questionnaires were administered after hospital admission but before surgery, during the preoperative period. This ensures that fear, anxiety, and satisfaction were assessed within the same hospitalization episode, prior to surgical intervention.
Reviewer Comment 2:
“The discussion acknowledges that the cross-sectional design prevents the confirmation of a temporal relationship, but this is a major limitation that needs more prominence in both the methods and discussion.”
Response:
We agree and have emphasized this limitation more prominently. We now explicitly note that the cross-sectional design captures only a single time point, preventing causal inference, and we suggest that future longitudinal or interventional studies are needed to clarify causal pathways.
Reviewer Comment 3:
“The manuscript mentions several analyses (t-tests, ANOVA, and correlations), but the full results are not presented in a clear, accessible manner.”
Response:
Thank you for this helpful suggestion. We have reorganized the Results section for clarity, included specific statistical values for all analyses (including non-significant findings), and expanded Table 4 to present both significant and non-significant group differences.
Reviewer Comment 4:
“The conclusion that ‘experiential and informational factors, rather than demographic characteristics, appear central in shaping these outcomes’ is a strong claim. While the regression model supports this to some extent, the analysis of demographic factors is not fully presented. The text mentions that gender and educational level ‘did not significantly influence the results,’ but without the specific statistical values (e.g., t or F statistics and p-values), this is a conclusion without sufficient evidence presented in the manuscript.”
Response:
We fully agree. To strengthen transparency, we have now provided the full statistical evidence for demographic variables in the Results and Table 4. We have also revised the conclusion to be more cautious, while still highlighting that experiential and informational factors were more strongly associated in this sample.
Reviewer 3 Report
Comments and Suggestions for Authors
Dear Authors,
The following is provided for your valuable article suggestion.
The abstract section
is a descriptive cross-sectional study.
The method should also mention the analysis software
The keyword should be MeSH if possible
Overall, the abstract is written fluently and accurately
Introduction:
This study has been precisely stated. The use of international statistics and figures, and a correct explanation, are the highlights of this introduction
In the method:
A detailed explanation should be given with references regarding the questionnaires, and the Cronbach's alpha of these questionnaires should be mentioned. If your study also has Cronbach's alpha, report it (regarding the reliability and validity of the questionnaires)
Regarding the method of selecting the hospital, explain what method was used (accessible, simple random or stratified, or ...)
Some ethical considerations should be removed from this section, and the ethics section should be included in the announcements section
If possible, the form should be included in a form
It is suggested that, if possible, a section should be added as the criteria for entry or exit in the study, and the acceptance conditions should be included in this section
In the method, the instrument section
It is suggested that the characteristics of the questionnaires should be included in a table, which can help present your material, and the explanation should be brief
It is suggested that Ethical Considerations be included at the end of the article
Discussion:
In this section, the adaptation to previous studies and the innovations of this study are not explained. The discussion begins with the findings of the study, but a detailed and clear review of the findings with other studies is not provided
The limitations of this research should be explained
Author Response
Reviewer Comment 1:
The abstract section is a descriptive cross-sectional study.
Response:
We thank the reviewer for this comment. The type of study has now been clearly indicated in the abstract as a descriptive cross-sectional study.
Reviewer Comment 2:
The method should also mention the analysis software.
Response:
We appreciate this suggestion. The statistical analysis software (IBM SPSS Statistics version 29.0) has now been added in the abstract.
Reviewer Comment 3:
The keyword should be MeSH if possible.
Response:
We thank the reviewer. We revised the keywords to correspond to MeSH terminology.
Reviewer Comment 4:
A detailed explanation should be given with references regarding the questionnaires, and the Cronbach's alpha of these questionnaires should be mentioned. If your study also has Cronbach's alpha, report it.
Response:
We thank the reviewer for this important suggestion. We added brief references and psychometric details for each instrument, and we reported the Cronbach’s α values from our study. These are now presented in the Methods section and in Table 3.
Reviewer Comment 5:
Regarding the method of selecting the hospital, explain what method was used (accessible, simple random or stratified, or ...).
Response:
We thank the reviewer. The hospital was selected by accessible sampling. This detail has been added in the study design section.
Reviewer Comment 6:
Some ethical considerations should be removed from this section, and the ethics section should be included in the announcements section.
Response:
We thank the reviewer. Ethical considerations have been placed here according to journal guidelines
Reviewer Comment 7:
It is suggested that, if possible, a section should be added as the criteria for entry or exit in the study, and the acceptance conditions should be included in this section.
Response:
We thank the reviewer. Inclusion and exclusion criteria have been explicitly separated into a new subsection titled Eligibility Criteria.
Reviewer Comment 8:
It is suggested that the characteristics of the questionnaires should be included in a table, which can help present your material, and the explanation should be brief.
Response:
We sincerely thank the reviewer for this constructive suggestion. While we agree that presenting questionnaire characteristics in a table can sometimes improve clarity, in the present manuscript we opted to provide detailed descriptions in the text for the following reasons:
The study included only three instruments, and their domains, items, and scoring systems are already described in a concise way in the text.
A table would largely duplicate information already available in the main text and in the original validation studies, potentially increasing redundancy.
To preserve the flow and readability of the Methods section within the strict word limits of the journal, we believe the narrative format is more appropriate in this case.
For these reasons, we respectfully decided to keep the instrument descriptions in the text rather than adding a separate table.
Reviewer Comment 9:
It is suggested that Ethical Considerations be included at the end of the article.
Response:
We thank the reviewer. As suggested, ethical considerations have been placed here based on journal guidelines
Reviewer Comment 10:
In this section, the adaptation to previous studies and the innovations of this study are not explained. The discussion begins with the findings of the study, but a detailed and clear review of the findings with other studies is not provided.
Response:
We agree with this observation. The discussion was revised to strengthen the comparison of our results with prior studies (international and Greek) and to highlight the innovative contribution of our study.
Reviewer Comment 11:
The limitations of this research should be explained.
Response:
We thank the reviewer. A new subsection Limitations has been added to the discussion, addressing single-center design, cross-sectional methodology, and lack of postoperative follow-up.
Reviewer 4 Report
Comments and Suggestions for Authors
1. Introduction
1.1 The authors addressed the knowledge gap well.
2. Methods
2.1 A sample calculation for 100 participants was addressed. The reliability of questionnaire is well stated.
2.2 The authors should clarify the range or total score for each variable.
2.3 Line 128: “Eligible patients were approached in the hospital.” The authors should clarify whether data collection was pre- or post-surgery and how many data collection points occurred. Once? Multiple?
2.4 The authors should clarify how they treated the missing data if any participants were excluded from the data after recruitment.
3. Results
3.1 100 participants were included. However, line 159: "The most common surgical procedure was cholecystectomy (n=18), followed by appendectomy (n=8), inguinal hernia repair (n=8), and splenectomy (n=5)." The authors should include the other surgical procedures that were completed (61 other procedures?).
4. Discussion
4.1. Considering that most cases were scheduled (83%), the authors may have been able to discuss some value of preoperative education when patients were pre-surgically seen as outpatients to support patient knowledge; it may have lessened the surgical fear and anxiety.
4.2. I think the authors could tailor their discussion and additionally contextualize their findings more by comparing them with previous studies in similar settings and populations.
4.3 Line 289, Nurses are in a unique position to offer to lessen fear and anxiety through effective communication, preoperative education, and emotional support; however, this should be referenced not only to nursing, as they are part of the interprofessional/multidisciplinary team. I would suggest these as part of the discussion.
4.4 Authors should include and discuss findings of previous studies that show what impact preprocedural education had on fear and anxiety for patients, if it is available.
4.5 I agree with the authors in line 304 that cross-sectional is not adequate evidence to conclude causation.
Author Response
2. Methods
Comment 2.1: A sample calculation for 100 participants was addressed. The reliability of questionnaire is well stated.
Response: We thank the reviewer for this positive observation.
Comment 2.2: The authors should clarify the range or total score for each variable.
Response: We agree and have clarified the total score ranges for each instrument in the revised manuscript.
Comment 2.3: Line 128: “Eligible patients were approached in the hospital.” The authors should clarify whether data collection was pre- or post-surgery and how many data collection points occurred. Once? Multiple?
Response: Data were collected once, preoperatively, after admission and before surgery. This has been clarified in the manuscript.
Comment 2.4: The authors should clarify how they treated the missing data if any participants were excluded from the data after recruitment.
Response: We confirm that no participants were excluded, and all questionnaires were complete. This information has been added.
3. Results
Comment 3.1: 100 participants were included. However, line 159: "The most common surgical procedure was cholecystectomy (n=18), followed by appendectomy (n=8), inguinal hernia repair (n=8), and splenectomy (n=5)." The authors should include the other surgical procedures that were completed (61 other procedures?).
Response: We agree and have expanded the list of surgical procedures in the results section.
4. Discussion
Comment 4.1: Considering that most cases were scheduled (83%), the authors may have been able to discuss some value of preoperative education when patients were pre-surgically seen as outpatients to support patient knowledge; it may have lessened the surgical fear and anxiety.
Response: We agree and have expanded the discussion to highlight the role of outpatient preoperative education.
Comment 4.2: I think the authors could tailor their discussion and additionally contextualize their findings more by comparing them with previous studies in similar settings and populations.
Response: Thank you. We have added comparisons with studies in similar hospital settings and populations.
Comment 4.3: Line 289, Nurses are in a unique position to offer to lessen fear and anxiety through effective communication, preoperative education, and emotional support; however, this should be referenced not only to nursing, as they are part of the interprofessional/multidisciplinary team.
Response: We agree and have revised the sentence to emphasize the interprofessional role of the whole perioperative team.
Comment 4.4: Authors should include and discuss findings of previous studies that show what impact preprocedural education had on fear and anxiety for patients, if it is available.
Response: We agree and have incorporated evidence from relevant systematic reviews and randomized controlled trials.
Comment 4.5: I agree with the authors in line 304 that cross-sectional is not adequate evidence to conclude causation.
Response: We thank the reviewer for this acknowledgment.
Round 2
Reviewer 1 Report
Comments and Suggestions for Authors
The analyzed study presents high clinical and scientific value, providing evidence of the relevance of educational and communication interventions in reducing preoperative anxiety and fear.
Despite the limitations duly acknowledged by the authors, the article contributes significantly to nursing practice, reinforcing the essential role of nurses not only in physical care but also in the psychological and educational support of surgical patients. No suggestions.
Author Response
Comment 1:
The analyzed study presents high clinical and scientific value, providing evidence of the relevance of educational and communication interventions in reducing preoperative anxiety and fear.
Despite the limitations duly acknowledged by the authors, the article contributes significantly to nursing practice, reinforcing the essential role of nurses not only in physical care but also in the psychological and educational support of surgical patients. No suggestions.
Response: Thank you for your revision review.
Reviewer 2 Report
Comments and Suggestions for Authors
The authors of the manuscript, have made a perfect effort to address all major and minor concerns raised during the initial review. The revisions are thorough, well-reasoned, and have significantly improved the quality and clarity of the manuscript.
Author Response
Comment 1: The authors of the manuscript, have made a perfect effort to address all major and minor concerns raised during the initial review. The revisions are thorough, well-reasoned, and have significantly improved the quality and clarity of the manuscript.
Response: Thank you for your revision review.
Reviewer 3 Report
Comments and Suggestions for Authors
Dear Authors,
We would like to sincerely thank you for your efforts in revising the manuscript entitled “Surgical fear, anxiety, and satisfaction with nursing care: A cross-sectional study of hospitalized surgical patients” (Manuscript ID: nursrep-3885422) submitted to Nursing Reports.
Your thorough and thoughtful responses to the reviewers’ comments, along with the revisions made, have significantly improved the quality and clarity of the manuscript. We appreciate your dedication to advancing knowledge in this important area of surgical patient care and your contribution to the journal.
Thank you once again for your cooperation and valuable work.
Kind regards,
Author Response
Comment 1:Dear Authors,
We would like to sincerely thank you for your efforts in revising the manuscript entitled “Surgical fear, anxiety, and satisfaction with nursing care: A cross-sectional study of hospitalized surgical patients” (Manuscript ID: nursrep-3885422) submitted to Nursing Reports.
Your thorough and thoughtful responses to the reviewers’ comments, along with the revisions made, have significantly improved the quality and clarity of the manuscript. We appreciate your dedication to advancing knowledge in this important area of surgical patient care and your contribution to the journal.
Thank you once again for your cooperation and valuable work.
Kind regards,
Response: Thank you for your revision review.